# Deep generative models for T cell receptor protein sequences

**Kristian Davidsen[1,2]\*, Branden J Olson[1,2], William S DeWitt III[1,2], Jean Feng[1,2], Elias Harkins[1,2], Philip Bradley[1,2], Frederick A Matsen IV[1,2]\***

[1]University of Washington, Seattle, United States; [2]Fred Hutchinson Cancer Research Center, Seattle, United States

**Abstract** Probabilistic models of adaptive immune repertoire sequence distributions can be used to infer the expansion of immune cells in response to stimulus, differentiate genetic from environmental factors that determine repertoire sharing, and evaluate the suitability of various target immune sequences for stimulation via vaccination. Classically, these models are defined in terms of a probabilistic V(D)J recombination model which is sometimes combined with a selection model. In this paper we take a different approach, fitting variational autoencoder (VAE) models parameterized by deep neural networks to T cell receptor (TCR) repertoires. We show that simple VAE models can perform accurate cohort frequency estimation, learn the rules of VDJ recombination, and generalize well to unseen sequences. Further, we demonstrate that VAE-like models can distinguish between real sequences and sequences generated according to a recombination-selection model, and that many characteristics of VAE-generated sequences are similar to those of real sequences.
DOI: https://doi.org/10.7554/eLife.46935.001

## Introduction

T cell receptors (TCRs) are composed of an $\alpha$ and a $\beta$ protein chain, both originating from a random V(D)J recombination process, followed by selective steps that ensure functionality and limit auto-reactivity. To generate diverse and functional TCRs, T cells combine a stochastic process for choosing from a pool of V, D and J genes with a process for selecting for expression and MHC recognition. The process first occurs for the $\beta$ chain, where first a D and a J gene are recombined using random trimming and joining with random nucleotides, then this DJ segment is recombined with a V gene via an analogous process. After the $\beta$ chain has been generated, a small cell expansion occurs followed by a similar $\alpha$ chain recombination, although without a D gene. For detailed reviews of V(D)J recombination see *Bassing et al. (2002)*, and *Schatz and Ji (2011)*. The naive T cell population consists of T cells that have undergone V(D)J recombination and MHC selection but not yet encountered antigen. In a system known as the clonal selection mechanism of immune memory, T cells that bind antigen increase in frequency, thus increasing the frequency of their corresponding TCR sequences. The resulting ensemble of protein sequences thus summarizes each individual's previous immune exposures and largely determines their resistance to various infections. One can consider these protein sequences as a sample from a probability distribution, whether it is the distribution of receptors within an individual, or the distribution of receptors in a population. This article concerns fitting such probability distributions on TCR $\beta$ protein sequences (which will be called 'TCR sequences' for the rest of the paper).

Probability estimates from these models can be used to draw important biological conclusions. For example, observing sequences that are amplified in a repertoire indicates that they perform important functions like targeting yellow fewer or cytomegalovirus (*Pogorelyy et al., 2018c*; *Pogorelyy et al., 2018d*; *Emerson et al., 2017*). However, in order to properly define amplification,

**\*For correspondence:**
krdav@uw.edu (KD);
matsen@fredhutch.org (FAMIV)

**Competing interests:** The authors declare that no competing interests exist.

we must infer the frequency of such sequences appearing in the naive (i.e. post-selection but pre-amplification) repertoire so that we do not mistake an inherently probable recombination scenario with functional selection. As another application, (*Elhanati et al., 2018*) used probability calculations to predict the frequency of shared TCR sequences between individuals, showing that biases of the V (D)J recombination process significantly explain the degree of sharing.

The appearance of a given TCR sequence in the blood of an individual means that it was generated by V(D)J recombination and subsequently passed thymic selection, which removes TCRs with improper binding to MHC as well as self-reactive TCRs. This series of two steps constitutes a sophisticated random process for generating protein sequences. Previous work (*Elhanati et al., 2014*; *Pogorelyy et al., 2018c*) approached the problem of inferring this process by calculating the probability of a sequence's V(D)J recombination using a probabilistic graphical model, multiplying this probability by a thymic selection factor $Q$, and scaling accordingly. Although breaking the process into generation and selection steps parallels the biological process, we can instead fit a distribution to a mature TCR repertoire directly, and assess the advantages of either approach. Indeed, these considerations raise the question of how to model the distribution of TCR protein sequences from a given source in order to answer meaningful immunological questions.

In this paper we develop variants of Variational Autoencoder (VAE) models (*Kingma et al., 2014b*; *Higgins et al., 2017*) to fit the distribution of TCR protein sequences. Recent work on deep generative models of proteins inspired our approach (*Sinai et al., 2017*; *Riesselman et al., 2018*). We find that these models can predict cohort frequency with high accuracy, learn the rules of VDJ recombination, generalize to unseen sequences, and generate sequences with similar characteristics to observed TCR sequences.

## Results

### Methods overview

We briefly outline our methods in order to present results; further details can be found in the Materials and methods section. We model TCR sequences using simple variants of variational autoencoders (VAEs). Previous work using VAEs have found success when first, there is a vast amount of data available, and second, the data distribution is complicated, involving nonlinearities and interactions between covariates. There is indeed a vast amount of TCR repertoire data, and the TCR probability distributions are complex.

VAE models can be described as consisting of an $n$-dimensional latent space, a prior $p_\theta(\mathbf{z})$ on that latent space, and probabilistic maps parameterized by two neural networks: an encoder $q_\phi(\mathbf{z}|\mathbf{x})$ and a decoder $p_\theta(\hat{\mathbf{x}}|\mathbf{z})$ (*Figure 1*; *Kingma et al., 2014b*). For the models used in this paper the latent space is 20-dimensional, and we use the conventional choice of a standard multivariate normal prior for $p_\theta(\mathbf{z})$. The encoder $q_\phi(\mathbf{z}|\mathbf{x})$ is a multivariate normal distribution with mean and diagonal covariance determined by a neural network with input $\mathbf{x}$ (see Materials and methods for how TCR protein sequences are transformed into appropriate input for a neural network). This choice of a normal distribution is primarily for mathematical convenience rather than being part of a specific modeling design; the normal 'noise' in the latent space gets processed by a neural network which introduces non-linearities that ensure that the result is not normal. However, VAE variants do use other distributions in place of normal (*Dilokthanakul et al., 2016*; *Davidson et al., 2018*). The decoder $p_\theta(\hat{\mathbf{x}}|\mathbf{z})$ is a per-site categorical distribution over amino acids and gaps parameterized by a neural network with input $\mathbf{z}$.

Once the VAE is trained, one can sample new sequences by 'decoding' samples from $p_\theta(\mathbf{z})$, that is, drawing from $p_\theta(\mathbf{z})$, feeding those points through the decoder network, and then sampling from the resulting probabilities. In the case of TCRs, this final sampling step goes from categorical distributions on the TCR components (i.e. on the V gene, J gene, and the amino acids at the various positions) to an actual TCR sequence. One trains a VAE with a collection of observed sequences $\mathbf{x}$ via the encoder $q_\phi(\mathbf{z}|\mathbf{x})$. VAE training has two goals, which are represented by two terms of the objective function: first, to be able to (probabilistically) encode and decode the sequences through the latent space with high fidelity, and second, to ensure that that the $q_\phi(\mathbf{z}|\mathbf{x})$ map is close to the prior $p_\theta(\mathbf{z})$ on average across $\mathbf{x}$. The second component of this objective encourages a structured mapping of input sequences to latent values, in hopes that the model learns meaningful sequence

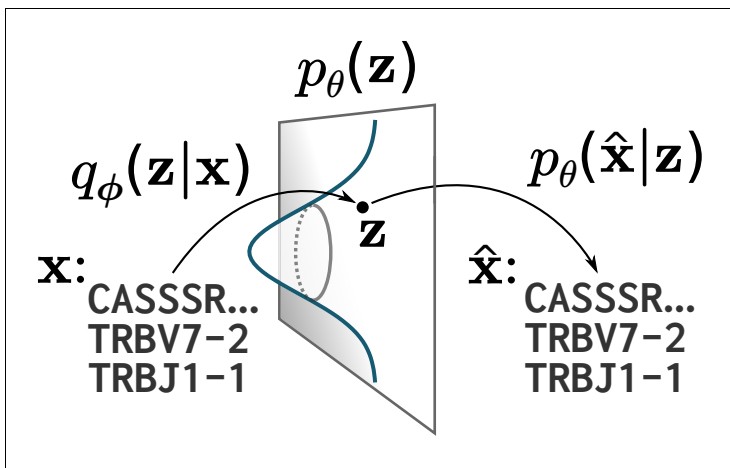

**Figure 1.** A cartoon of a variational autoencoder (VAE). A VAE embeds objects of interest $\mathbf{x}$ (here TCR protein sequences) into an $n$-dimensional latent space, using a probabilistic encoder $q_\phi(\mathbf{z}|\mathbf{x})$ and decoder $p_\theta(\hat{\mathbf{x}}|\mathbf{z})$ that are both parametrized by deep neural networks. The VAE objective is to encode and decode objects with high fidelity ($\mathbf{x} \approx \hat{\mathbf{x}}$) while ensuring the encoder $q_\phi(\mathbf{z}|\mathbf{x})$ distribution is close to a prior $p_\theta(\mathbf{z})$ on that latent space, typically taken to be a standard multivariate normal distribution.

DOI: https://doi.org/10.7554/eLife.46935.016

The following figure supplements are available for figure 1:

**Figure supplement 1.** The `basic` model decoder.
DOI: https://doi.org/10.7554/eLife.46935.017

**Figure supplement 2.** The `count_match` model decoder.
DOI: https://doi.org/10.7554/eLife.46935.018

features rather than memorizing properties of the training data. The balance between these two components is important and is controlled by a parameter $\beta$ (*Higgins et al., 2017*). Once the VAE is trained (i.e. parameters $\phi$ and $\theta$ are optimized according to the objective with respect to a particular dataset), we can calculate the probability of generating a given sequence $\mathbf{x}$ via importance sampling.

We are interested in TCR $\beta$ protein sequences, which due to the process of VDJ recombination are uniquely identified by triples consisting of V gene, J gene, and CDR3 protein sequence (*Woodsworth et al., 2013*). We developed two VAE models for such protein sequences: a simple one, denoted `basic` and a more complex model, denoted `count_match`. The `basic` model does not have any information about the content of germline genes built into the model and was trained according to a simple loss function (*Figure 1—figure supplement 1*). The `count_match` model brings in information about the protein sequence of the germline genes and has a more complex loss function involving CDR3 length and the degree to which the protein sequences on the ends of the CDR3 match the corresponding germline gene sequences (*Figure 1—figure supplement 2*).

As a baseline for comparison, we combined OLGA, a sophisticated recombination model (*Sethna et al., 2018*), with a simplified version of the selection model used in *Elhanati et al. (2014)*, together which we will denote `OLGA.Q`. Our selection component $Q$ is parameterized by triples consisting of V gene identity, J gene identity, and CDR3 length, resulting in a model with about 14,000 parameters. This is a simpler model than the general *Elhanati et al. (2014)* model, which allows for selection based on CDR3 amino acid composition. However, it is a richer model than any models used by the same group since the publication of *Elhanati et al. (2014)*, such as the one used to find condition-associated immune receptors in *Pogorelyy et al. (2018b)* and *Pogorelyy et al. (2018c)*. *Sethna et al. (2018)* suggest probabilistically evaluating vaccine targets using OLGA directly and no selection model at all. In any case, a software implementation of the general *Elhanati et al. (2014)* model, for which training is highly involved, is not currently available.

## VAE models predict cohort frequency

We wished to understand the ability of `basic`, `count_match`, and `OLGA.Q` to estimate the frequency with which a TCR appears in a given cohort, both when the TCR is contained in the training set ('train') and when it is not ('test'). Here we define 'cohort count' for a collection of repertoires to be the number of times a given TCR amino acid sequence appeared in the output files from the ImmunoSEQ assay (Adaptive Biotechnologies, Seattle, WA, USA) for those repertoires (ignoring the template abundance column). Multiple nucleotide occurrences of a given TCR protein sequence contribute separately to this number. Define $c$ to be the cohort count vector for the data set of *Emerson et al. (2017)*, indexed by the TCR protein sequences with values being these cohort counts.

To assess out-of-sample performance, we first partitioned $c$ into $c_{\text{train}}$ and $c_{\text{test}}$ with a 50/50 split irrespective of abundance. We emphasize that there is no overlap between these collections of TCRs. To obtain a training set, we drew 200,000 sequences from the multinomial distribution induced by $c_{\text{train}}$. We then trained `basic` and `OLGA.Q` using these sequences. The trained models yield per-sequence probability distributions: $P_{\text{VAE}}$ for the `basic` VAE and $P_{\text{OLGA.Q}}$ for `OLGA.Q`. We evaluated each of these probabilities as well as the cohort frequency for 10,000 sequences drawn multinomially from $c_{\text{test}}$ or $c_{\text{train}}$.

We performed this procedure for the entire cohort, but also restricting the cohort for training to a randomly-selected, smaller number of subjects while still comparing to frequency estimates using the whole cohort.

We found that VAE models can predict cohort frequency for out-of-sample TCR sequences (*Figure 2*). As the number of samples increases, the scatter of points decreases and the difference between training and testing samples also decreases. With 666 samples, this results in an $R^2$ value

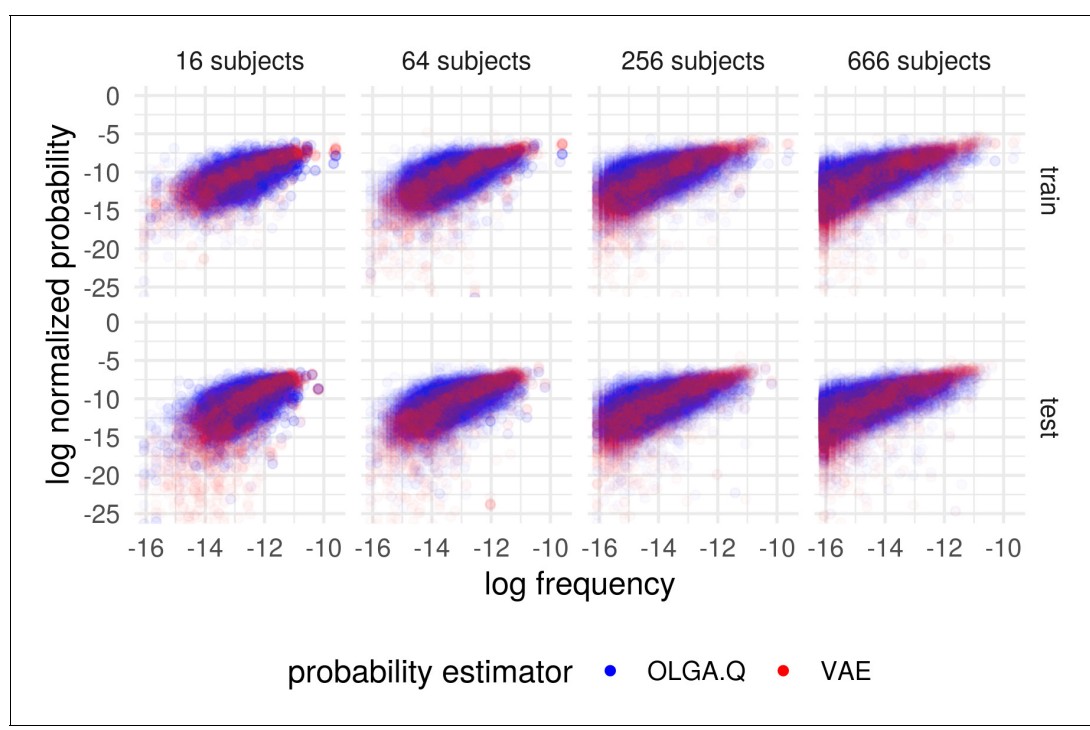

**Figure 2.** Cohort frequency prediction with two probability estimators. Plot shows the (natural) log frequency in the entire cohort, restricted to TCRs appearing in the subset of subjects, versus the probability according to $P_{\text{OLGA.Q}}$ and $P_{\text{VAE}}$ for the `basic` model. Results partitioned into when the TCR appeared in the training set ('train') and when it did not ('test'). Probabilities for each estimator normalized to sum to one across the collection of sequences represented in the plots.

DOI: https://doi.org/10.7554/eLife.46935.002

The following figure supplement is available for figure 2:

**Figure supplement 1.** Comparison of $R^2$ values for cohort frequency estimation.

DOI: https://doi.org/10.7554/eLife.46935.003

for the best-fit line on the log-log scale of 0.258 for $P_{\mathrm{OLGA.Q}}$ and an $R^2$ value of 0.442 for $P_{\mathrm{VAE}}$ on the test set (*Figure 2—figure supplement 1*). When we increased the number of training sequences five-fold to 1 million, $R^2$s increased slightly to 0.268 for $P_{\mathrm{OLGA.Q}}$ and 0.474 for $P_{\mathrm{VAE}}$. Recall that these correlation measures include the full scale of frequencies, including very noisy frequency estimates on the lower end of the scale. Also, we make no efforts to account for sequencing error above the methods used in *DeWitt et al. (2018)*. We note that higher correlations have been observed for an `OLGA.Q`-type model when calculating probability of CDR3 only, restricting to sequences found in an epitope database, and smoothing using a single amino acid mismatch (*Pogorelyy et al., 2018a*).

## VAE models learn the rules of VDJ recombination

TCR $\beta$ chains are generated via VDJ recombination, a process in which germline-encoded genes are randomly chosen from a pool, trimmed a random amount, and then joined together with random nucleotide insertions in between. This recombination process leads to important structural characteristics in the generated sequences. Specifically, because the beginning of the CDR3 region is encoded by the V gene, and the end by the J gene, there is a strong correlation between the V and J gene identities and the CDR3 sequence.

The probabilistic models considered here differ in the extent to which they explicitly model this process. On one end of the spectrum, the `OLGA.Q` model is built on an explicit model of nucleotide VDJ recombination which emulates this process quite carefully, using our knowledge of the germline TCR nucleotide sequences and recombination mechanism (*Murugan et al., 2012*; *Marcou et al., 2018*; *Sethna et al., 2018*). The `count_match` model incorporates some of these aspects by making the germline V and J amino acid sequences for each input available to the decoder, and by scoring the degree to which the correct number of CDR3 amino acid positions of the reconstructed sequences match those of their corresponding V and J genes. The `basic` model predicts the germline genes and the CDR3 sequences as independent outputs of the VAE, and thus has no built-in prior information on the correlations between the germline genes and CDR3s.

We can understand the degree to which the models learn the VDJ recombination rules by evaluating them under the $P_{\mathrm{OLGA.Q}}$ recombination-selection model. If the VAE models respect the rules of VDJ recombination, they will generate sequences with a $P_{\mathrm{OLGA.Q}}$ comparable to that of real sequences, while if they do not respect these rules, they should get a low $P_{\mathrm{OLGA.Q}}$. This is a stringent criterion: a single amino acid change towards the $3'$ end of the CDR3 can cause $P_{\mathrm{OLGA.Q}}$ to drop precipitously. For example, OLGA gives (TRBV5-1, TRBJ2-6, CASSFSGSGANVLTF) a relatively high probability, while the same TCR with a single T switched to Q (TRBV5-1, TRBJ2-6, CASSFSGS-GANVL**Q**F) is assigned probability zero.

To test the models' compliance with the rules of VDJ recombination, we used the data of *De Neuter et al. (2019)*, which consists of TCR $\beta$ sequences from 33 subjects, as follows. We randomly split the data so that the repertoires of 22 subjects were used for training, and the remaining 11 subjects' repertoires were used for testing (with one repertoire used for each subject). Each of the 22 training repertoires was randomly downsampled to 20,000 sequences to standardize the contribution of each repertoire to the training set; these samples were then pooled. 100,000 sequences from this pool were randomly selected to train the models, including the $Q$ factor of `OLGA.Q`. We then evaluated the distribution of $P_{\mathrm{OLGA.Q}}$ on 10,000 sequences from each of the held-out test repertoires as well as 10,000 sequences generated from each of the three models.

We found evidence that the VAE models do indeed learn the rules of VDJ recombination (*Figure 3*). Although there is slight left skew in the $P_{\mathrm{OLGA.Q}}$ distributions for VAE-generated sequences compared to the $P_{\mathrm{OLGA.Q}}$ distributions of experimental repertoires, the behavior of the VAE-generated distributions reasonably matches the behavior of the experimental distributions. In fact, the $P_{\mathrm{OLGA.Q}}$ distribution for `OLGA.Q`-generated sequences seems to exhibit more left skew than the $P_{\mathrm{OLGA.Q}}$ distributions for VAE-generated sequences, although the three distributions are for the most part comparable. Perhaps surprisingly, the `count_match` model that encodes germline amino acid information resulted in a very small improvement in terms of recombination probability compared to the `basic` model, which does not explicitly encode any dependence between a TCR's germline gene usage and CDR3 sequence.

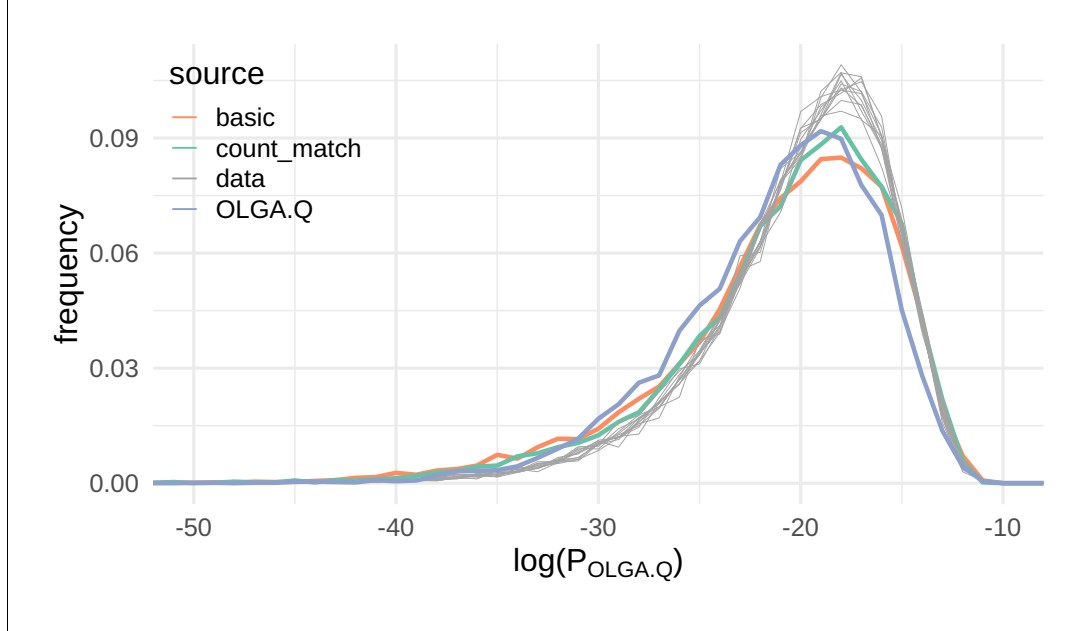

**Figure 3.** VAE models generate plausible recombinations according to the `OLGA.Q` model, which is built on a model of VDJ recombination. Here we show the distribution of log-probability of generation according to the `OLGA.Q` model for a panel of sequences from 11 test repertoires (gray) as well as simulated sequences from the `basic`, `count_match`, and `OLGA.Q` models.

DOI: https://doi.org/10.7554/eLife.46935.004

## VAE models generalize to unseen sequences and learn more than a simple `OLGA.Q`

Next, we set out to determine whether the VAE models were simply memorizing and regurgitating training sequences. Such behavior is a persistent concern for deep generative models (*Arora et al., 2017*; *Arora and Zhang, 2017*). Although the close correspondence between test and train performance in the above frequency estimation suggests model generalization, it does not directly address this issue.

To evaluate out-of-sample probability estimation, we used the *De Neuter et al. (2019)* data as in the previous section to evaluate $P_{\mathrm{VAE}}$ under the `basic` model rather than $P_{\mathrm{OLGA.Q}}$. If the VAE were regurgitating training sequences, it should consistently assign higher $P_{\mathrm{VAE}}$ to sequences it generates compared to held-out test sequences. Instead, we found that the $P_{\mathrm{VAE}}$ probabilities for VAE-generated sequences closely follow probabilities for test sequences (*Figure 4*), for both basic and `count_match`. We also observed that the `OLGA.Q`-generated sequences are consistently assigned a lower $P_{\mathrm{VAE}}$ on average than either test sequences or VAE-generated sequences, indicating the VAE learns characteristics of real sequences not captured by the formulation of `OLGA.Q` used here.

## VAE models generate sequences with similar characteristics to real sequences

We next sought to quantify the similarity of model-generated sequences to real sequences, for each of the three models in consideration. To accomplish this task, we used the sumrep package (*Olson et al., 2019*) (https://github.com/matsengrp/sumrep/), a collaborative effort of the AIRR (*Breden et al., 2017*; *Rubelt et al., 2017*) software working group. This package calculates many summary statistics on immune receptor sequence repertoires and provides functions for comparing these summaries. While these summaries are not of direct interest for this application, they comprise simple and relevant means of summarizing the abstract, high-dimensional distribution of TCRs. Collectively, these summary comparisons allow for robust model validation without appealing to the models themselves for assessment. We found agreement between simulated and test repertoires in some respects, with the performance of the model depending on the summary statistic (*Figure 5*).

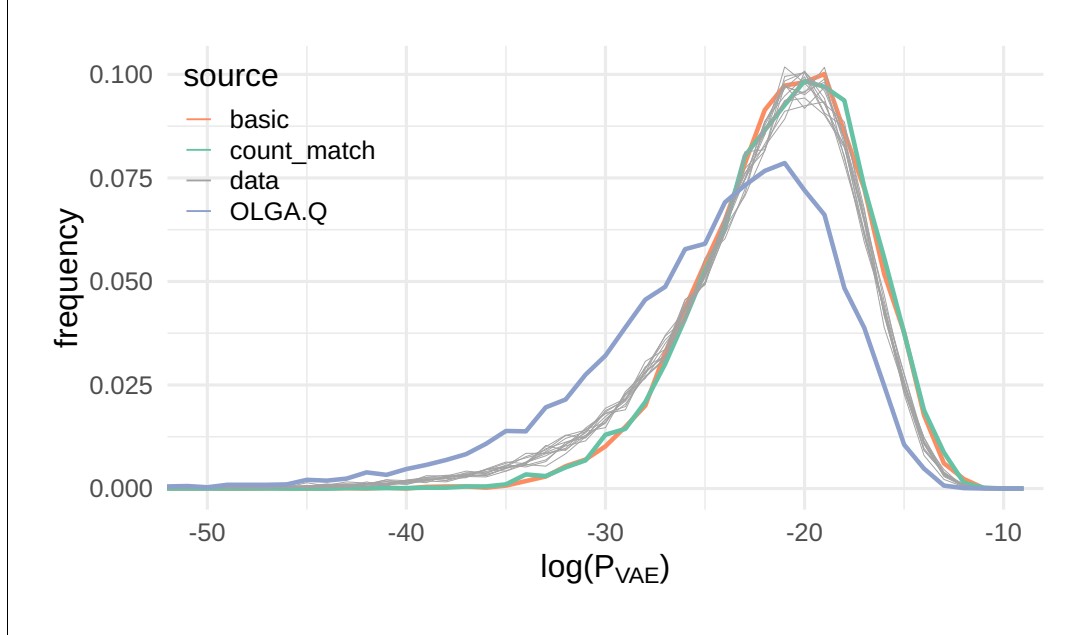

**Figure 4.** Sequences generated by the VAE models show a similar distribution of $P_{VAE}$ compared to real sequences. Here we show the distribution of the log probability of generation according to the OLGA.Q model for a panel of sequences from 11 test repertoires (gray) as well as simulated sequences from the basic, count_match, and OLGA.Q models.

DOI: https://doi.org/10.7554/eLife.46935.005

Above we showed how the VAE model learns the rules of VDJ recombination assessed by OLGA.Q likelihood distributions. Another assessment is to look directly at summary statistics like V/J gene usage, amino acid frequencies and CDR3 length. All models succeeded on J gene frequencies, with the VAE models performing worse in terms of V gene frequency, in particular the basic model. The models performed similarly in terms of CDR3 summaries, with OLGA.Q perhaps doing better in terms of CDR3 length, and the VAEs getting the correct distribution of nearest-neighbor distances (*Figure 5—figure supplement 1*). The amino acid frequencies for the VAE models did not match those of the training data as closely as expected, although in some respects they appear better than OLGA.Q. Results were broadly consistent when analyzing a second data set (*Figure 5—figure supplement 2*).

## The latent space embedding

We wished to understand the factors that determine the position of TCRs within the latent embedding. To uncover these determinants, we performed standard principal components analysis (PCA) on De Neuter test data embedded in the VAE latent space. This reduces the 20-dimensional latent space embedding to the two dimensions which account for the largest variability in the data.

We found that this projection is structured according to V and J gene identity (*Figure 6*). In particular, the V gene determines one axis of the principal components projection, while the J gene determines another. In order to learn the next level of organization, we restricted the embedded TCR sequences to those using the most popular V and J genes — TCRBV30-01 and TCRBJ01-02 — and re-did the projection. This additional projection showed that CDR3 length was an important determinant of embedding location (*Figure 6—figure supplement 1*).

## Discussion

Probabilistic models of immune repertoires are powerful tools, with applications to finding disease-responsive TCRs (*Pogorelyy et al., 2018c*) and analyzing the forces dictating TCR sharing (*Elhanati et al., 2018*), among others. In this paper we applied deep learning to model TCR $\beta$ repertoires, and used the resultant models to gain meaningful insights. Specifically, we use a

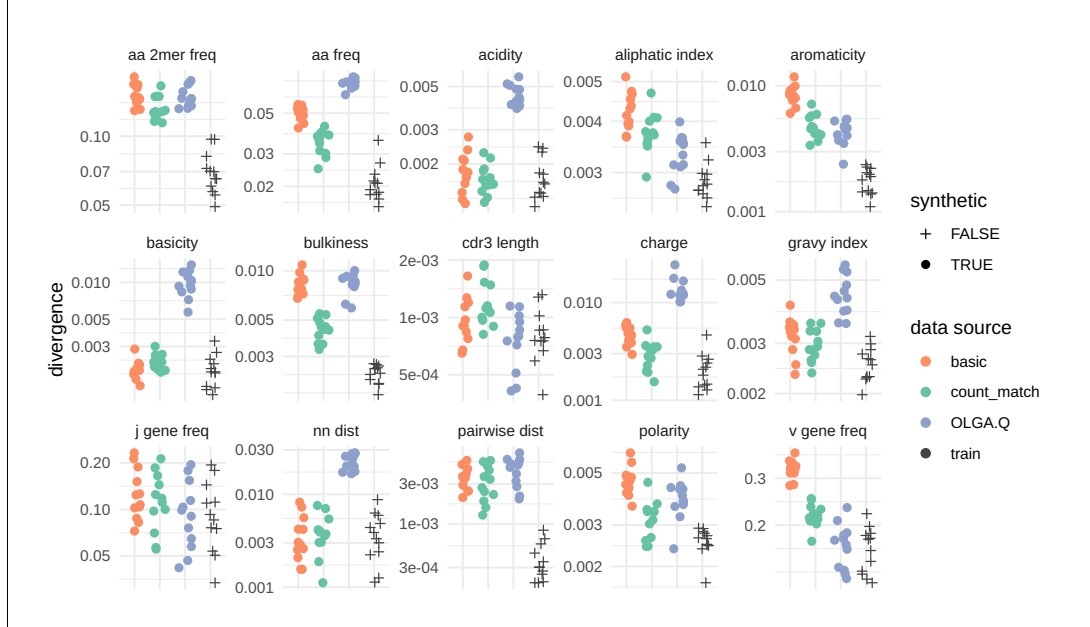

**Figure 5.** Divergences for summary statistics comparing model-generated sequences to held-out repertoire sequences on the *De Neuter et al. (2019)* data set. Each colored point represents the divergence of a summary distribution computed on a simulated pool of sequences to the distribution of the same summary on a set of sequences drawn from one of 11 repertoires (*Figure 5—figure supplement 1*). Each black '+' represents a similar divergence but with a random selection from the training data rather than a simulated pool of sequences. A lower divergence means more similarity with respect to the given summary. The following summary statistics, applied to the CDR3 amino acid sequence, use Jensen-Shannon divergence: acidity, aliphatic index, aromaticity, basicity, bulkiness, length (in amino acids), charge, GRAVY index, nearest neighbor Levenshtein distance, pairwise Levenshtein distance, and polarity. The following summary statistics use $\ell_1$ divergence: CDR3 amino acid 2mer frequency, CDR3 amino acid frequency, J gene frequency, and V gene frequency.

DOI: https://doi.org/10.7554/eLife.46935.006

The following figure supplements are available for figure 5:

**Figure supplement 1.** Nearest neighbor Levenshtein distributions on the *De Neuter et al. (2019)* data set.

DOI: https://doi.org/10.7554/eLife.46935.007

**Figure supplement 2.** Summary statistics comparison on a multiple sclerosis data set.

DOI: https://doi.org/10.7554/eLife.46935.008

semiparametric method that makes a single weak assumption: that there exists some small number of latent parameters that can be used to generate to the observed distribution. We make no assumptions about the function mapping from these parameters to the high-dimensional distribution space and learn it from the data. We have learned that this biology-agnostic approach can provide good results, even when compared to a previous approach that formalizes the considerable biological knowledge we have concerning the mechanism of VDJ recombination.

We find that these models have the following interesting features.

1. These models yield better in-sample and out-of-sample performance for cohort frequency estimation compared to an existing recombination and selection model.
2. They generalize well by learning features of real TCR repertoires, which allows them to differentiate between experimental repertoires and repertoires generated from the recombination and selection model.
3. They generate simulated repertoires that are similar to real TCR repertoires.
4. By leveraging powerful deep learning libraries, they can be expressed and implemented very simply with small amounts of specialized computer programming. The `basic` model, for example, is implemented in about 100 lines of Python code.

Furthermore, our efforts to inject biological knowledge into the deep learning framework did not significantly improve performance.

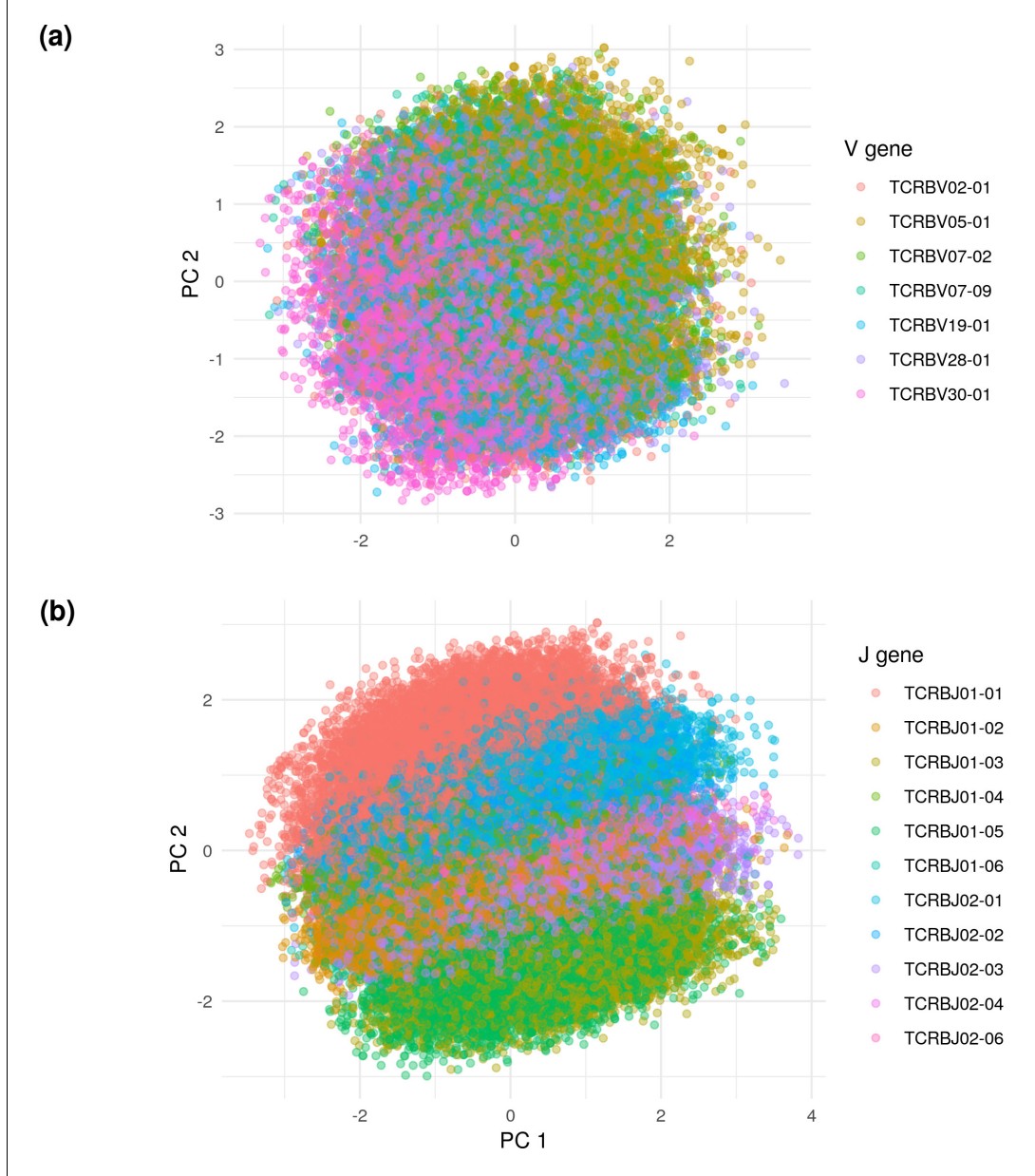

**Figure 6.** Principal components analysis (PCA) on the De Neuter test data embedded into the 20 dimensional latent space, colored by (a) V gene and (b) J gene. Panel (a) is limited to the seven most popular V genes.

DOI: https://doi.org/10.7554/eLife.46935.009

The following figure supplement is available for figure 6:

**Figure supplement 1.** PCA of the most popular V and J genes showing embedding of CDR3 length.

DOI: https://doi.org/10.7554/eLife.46935.010

However, these models also have some important drawbacks. Most importantly, as is often the case for models parametrized by neural networks, these models are not directly interpretable. Although we have identified some structure in the latent space, further details may be difficult to ascertain. Besides the difficulty in interpreting the neural network weights, we did not engineer the model architectures with mechanism in mind. In addition, the models operate on amino acid sequences and thus cannot shed light on the VDJ recombination process, which operates at the nucleotide level. We also note that $P_{\mathrm{VAE}}$, which relies on importance sampling, is more expensive to compute than $P_{\mathrm{OLGA.Q}}$.

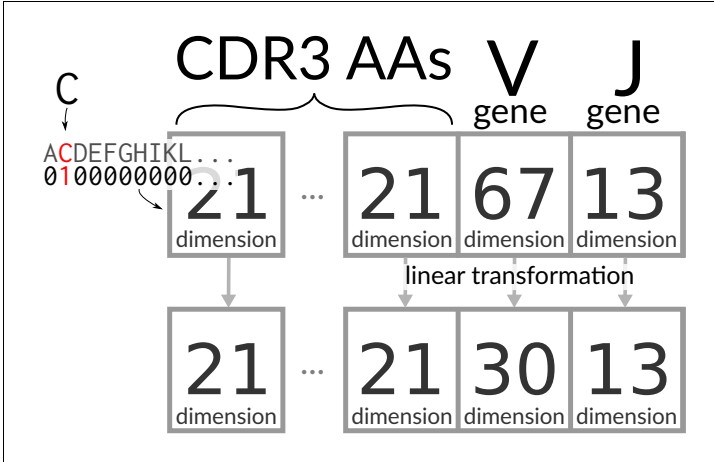

**Figure 7.** Encoding/transforming TCR sequences.
DOI: https://doi.org/10.7554/eLife.46935.011

The results presented here offer some interesting lessons concerning future development of deep probabilistic models for immune repertoires. Our model that had no a priori information about germline gene sequence performs very similarly, even when evaluated in terms of VDJ recombination likelihood, to one that deliberately attempts to recapitulate the amount of matching between germline gene and CDR3 amino acid sequences and includes germline CDR3 sequences in the untrained model. This may indicate that, given the volume of sequence data available, we should focus our efforts on the abstract problem of density estimation on the set of TCRs, rather than incorporating biological knowledge into our deep learning models.

Although we performed a preliminary analysis of the latent embedding, this exclusively involved sequence characteristics directly available to the model. In future work, we hope to further unravel this embedding by comparing repertoires in the latent space, and by comparing sequences labeled with external characteristics. We also plan to deliver a pre-trained model that will enable biologists to evaluate the probability of seeing a naive B cell receptor (BCR) or TCR in a given population. Here we have restricted our attention to TCR $\beta$ sequences, however, our methods apply with no modification to TCR $\alpha$ chains. Contrasting the $\alpha$ and $\beta$ chains may yield interesting insights on the differences between the two generation processes. The most interesting insights will come from jointly modeling the two chains using large-scale $\alpha\beta$ paired TCR sequencing (*Howie et al., 2015*), which is a more complex process.

## Materials and methods

### Data

Our goal was to model probability distributions on TCR $\beta$ chain protein sequences. By the process of VDJ recombination, these sequences are uniquely determined by V and J gene identities and CDR3 amino acid sequence. Thus, for the purpose of this paper, we exclusively used triples of V gene, J gene, CDR3 amino acid sequence to represent TCR protein sequences.

All data was downloaded from https://clients.adaptivebiotech.com/immuneaccess. We preprocessed the data to exclude sequences:

1. from an out-of-frame rearrangement
2. with a CDR3 that does not begin with the characteristic C or end with an F or YV
3. with a CDR3 longer than 30 amino acids
4. with an ambiguous V or J gene call.

We also excluded any TCRs with TCRBJ02-05, which the internal Adaptive pipeline annotates incorrectly, and TCRBJ02-07, to which the default OLGA model assigns artifactually low probabilities. Model design and parameter tuning, including the sizes of hidden layers and the dimension of

the latent space, was performed using the data of *DeWitt et al. (2018)*. On this data we endeavored to decrease model size without incurring loss on held-out data within this data set. We found that the model was relatively robust to parameter perturbations as long as the number of parameters was not too small. Model evaluation was performed using the data sets described in the Results section.

## Encoding TCR sequences

The CDR3 sequences were padded with gaps in the middle so that they are a fixed length of 30 amino acid/gap characters. Thus there is an equal number of amino acids on either end of the gaps for even length CDR3s, with one extra on the left side for odd length CDR3s. This resulting sequence is 'one-hot encoded,' meaning that each amino acid at each site is represented with 0/1 for absence/presence, with an additional dimension for 'gap' to make a 21-dimensional space (*Figure 7*). V and J genes are similarly encoded in 67- and 13-dimensional vectors, respectively, and all of these vectors are concatenated into a single large encoding vector. This vector is mapped to a latent embedding via a linear transformation that is learned during training (*Biswas et al., 2018*). In our case, there is one transformation for the V genes, one for the J genes, and one for amino acids. These transformations do not change dimension except for V gene identities, which are projected to a 30-dimensional space (*Figure 7*).

## Models

Here we describe the `basic` and `count_match` models in detail. They are not exactly VAEs as originally defined in *Kingma et al. (2014b)* for two reasons. First, they are better categorized as $\beta$-VAEs since they include a weight on the Kullback-Leibler divergence term of the training objective (*Higgins et al., 2017*). Namely, the loss is

$$\mathcal{R}(\mathbf{x}, \mathbf{z}) + \beta D_{\mathrm{KL}}(q_\phi(\mathbf{z}|\mathbf{x}) \| p_\theta(\mathbf{z})) \tag{1}$$

where $\mathcal{R}$ is the reconstruction loss for $\mathbf{x}$ encoded as $\mathbf{z}$ (details below).

Second, they have multiple outputs that are scored by separate reconstruction loss functions. Our reconstruction loss is a linear combination of these loss functions. For example, the simplest 'basic' model produces three outputs: one for the V gene, one for the J gene, and one for the CDR3 sequence. It has two densely-connected layers for the encoder and two for the decoder (*Figure 1—figure supplement 1*). The V and J gene identities are scored using categorical cross-entropy, while the CDR3 sequence is scored by the average categorical cross-entropy across sites.

The `count_match` model includes TCR germline information in the untrained model in such a way that it can count the number of V-germline-matching amino acids on the 5′ end of the CDR3 and the number of J-germline-matching amino acids on the 3′ end of the CDR3 (*Figure 1—figure supplement 2*). Its loss function includes a component that scores these counts in terms of two-dimensional squared loss. This model also contains an explicit loss component for CDR3 length, which is also evaluated via squared loss.

We combine the multiple losses within each model into a weighted linear combination which yields a single overall reconstruction loss function for optimization. Weights were determined by multivariate linear regression, minimizing the squared difference between the log likelihood and this reconstruction loss on a validation set (see next section for definition of the validation set used in training). This resulted in a marginal improvement in performance on the (*DeWitt et al., 2018*) data and fitting was not done again. Due to these modifications, our loss function cannot be interpreted in terms of the variational evidence lower bound (ELBO).

## Training

For the purposes of fitting, the training data was split into true-training and validation sets: the former was used for fitting, while the latter was used to assess error during training (which provided a stopping criterion). 'Test' data was completely held out from the training procedure.

Inspired by the work of *Sønderby et al. (2016)*, we implemented a $\beta$ schedule during training such that training begins with $\beta = 0$ and then linearly increases every training epoch until its final value. We extended this procedure by implementing a collection of pre-training phases that start with randomized weights and train for a fixed number of epochs using the $\beta$ schedule. The optimal

weights, according to the validation loss, were used as the starting weights for a full optimization, which terminates when validation loss does not improve for a fixed number of epochs or until a maximum number of epochs is reached. Training was done using the Adam optimizer *Kingma and Ba (2014a)* implemented in Keras (*Chollet, 2015*).

### Picking $\beta$

As described above, our complete loss function is a sum of a reconstruction loss plus $\beta$ times a Kullback-Leibler (KL) divergence term $D_{\mathrm{KL}}(q_\phi(\mathbf{z}|\mathbf{x})\|p_\theta(\mathbf{z}))$ describing the divergence of the probabilistic encoder map $q$ to the prior $p$. This KL divergence term regularizes the optimization by encouraging a structured embedding of TCRs.

We tested the $\beta$ parameter in an evenly distributed range with seven values from 0.625 to 1 on the *DeWitt et al. (2018)* data set. We found that $\beta$ slightly impacted $P_{\mathrm{VAE}}$ when evaluated on test sequences, with larger values of $\beta$ being slightly preferred (*Figure 8*). On the other hand, $\beta$ strongly impacted the agreement of summary statistics of generated sequences with observed sequences in test repertoires (*Figure 9*). To balance these evaluative and generative objectives, we fixed $\beta$ to be 0.75. This choice was confirmed by running the same analysis on the data of *Emerson et al. (2013)*

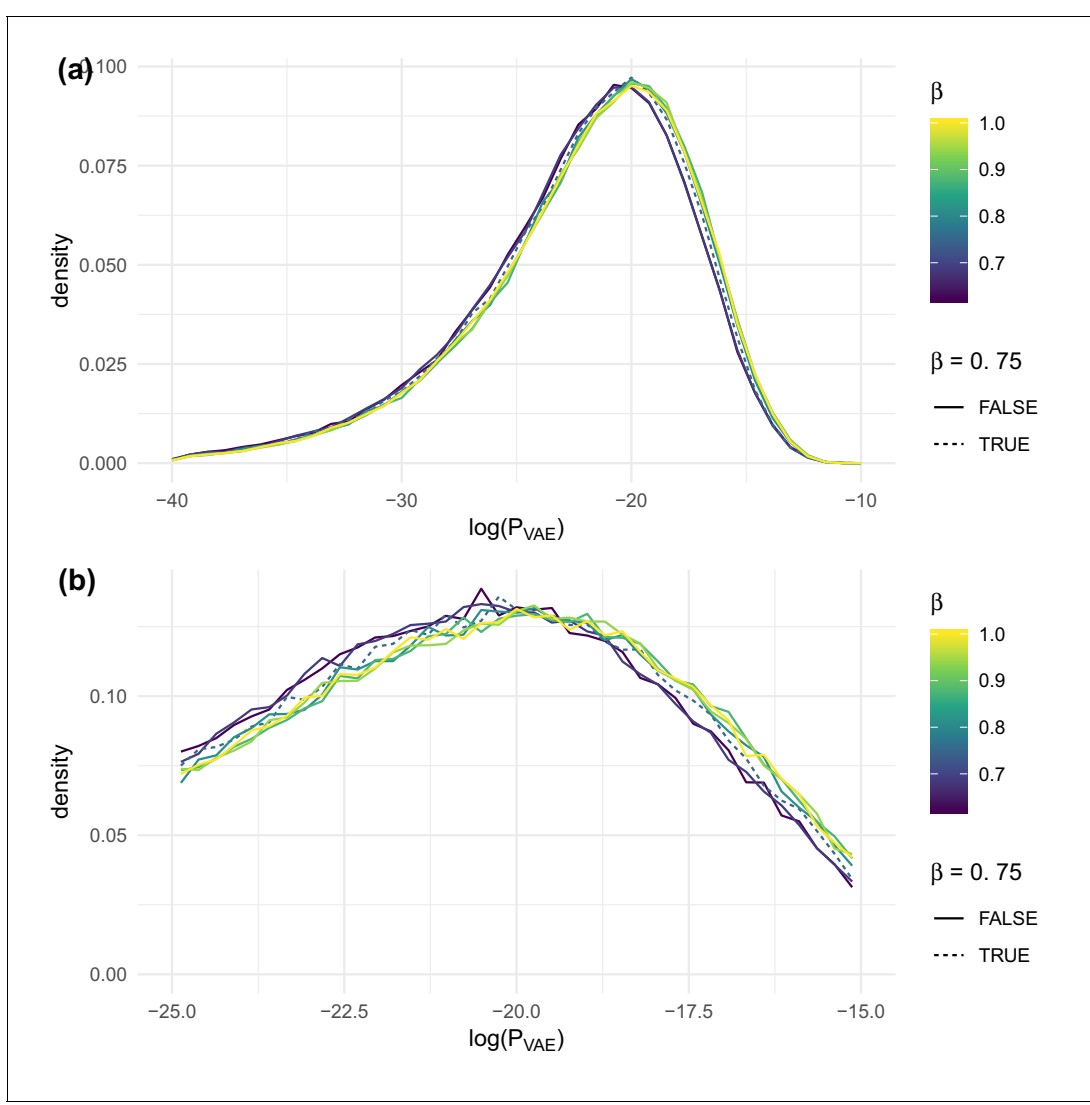

**Figure 8.** The effect of $\beta$ on $P_{\mathrm{VAE}}$ evaluated on test sequences for the data of *DeWitt et al. (2018)*, overall (a) and near the peak (b).
DOI: https://doi.org/10.7554/eLife.46935.012

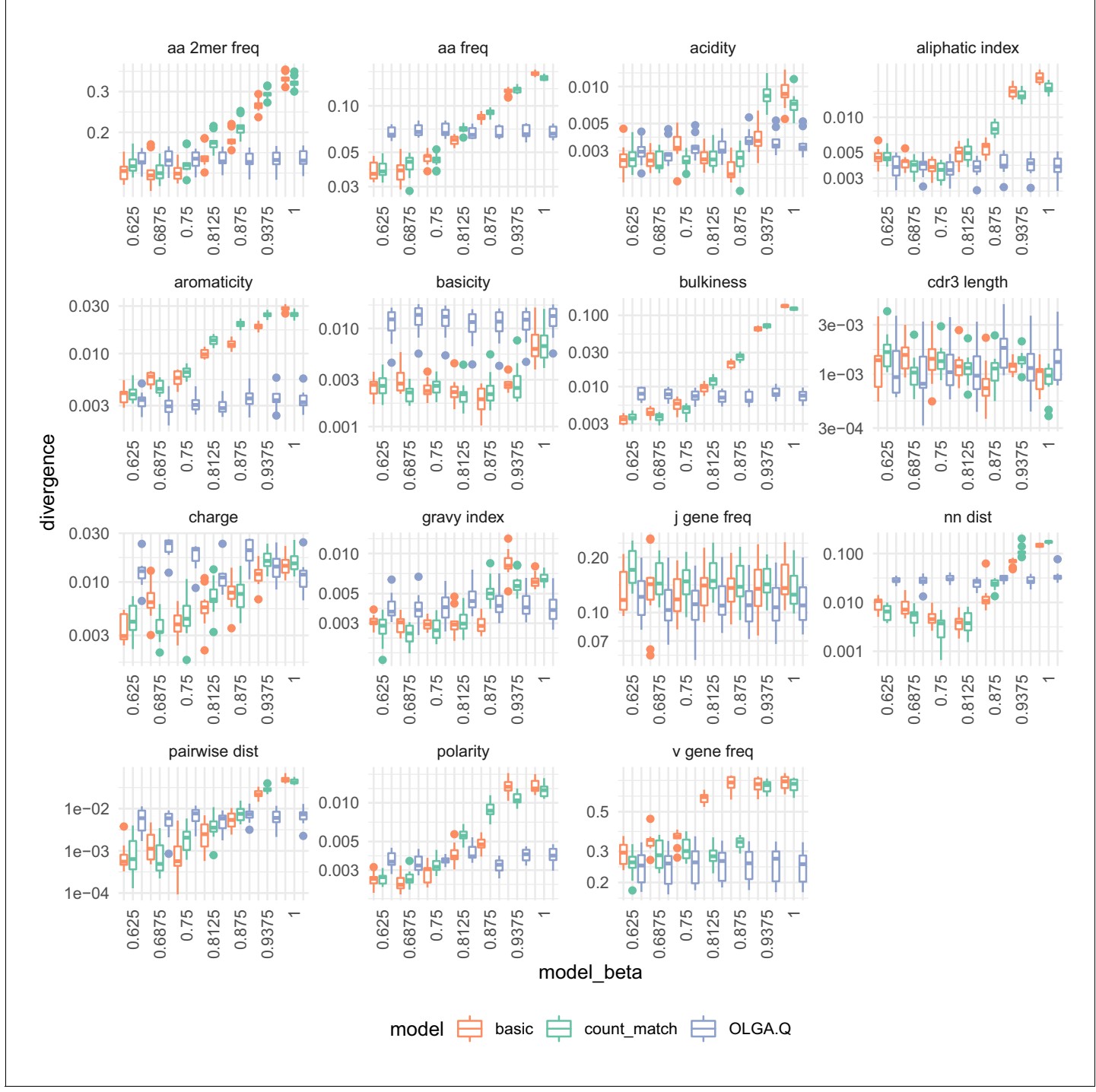

**Figure 9.** The effect of β on summary divergences between generated sequences and observed test sequences as in *Figure 4*, using the data of *DeWitt et al. (2018)*. OLGA.Q is also run separately for each β value; because β has no influence on OLGA.Q, the observed variation is simply due to differences between random samples.

DOI: https://doi.org/10.7554/eLife.46935.013

The following figure supplements are available for figure 9:

**Figure supplement 1.** Summary statistic divergences by β on the data of *Emerson et al. (2013)*.

DOI: https://doi.org/10.7554/eLife.46935.014

**Figure supplement 2.** Summary statistic divergences for each on the *De Neuter et al. (2019)*.

DOI: https://doi.org/10.7554/eLife.46935.015

(*Figure 9—figure supplement 1*) and *De Neuter et al. (2019)* (*Figure 9—figure supplement 2*), both of which yielded similar results.

## Importance sampling

$P_{\mathrm{VAE}}$ denotes the probability $p(\mathbf{x})$ of the VAE generating $\mathbf{x}$ when decoding a sample from the prior in the latent space. In principle we could calculate this as the expectation of $p(\mathbf{x}|\mathbf{z})$ where $\mathbf{z}$ is drawn from $p(\mathbf{z})$, but this would be very inefficient.

Instead, we use importance sampling, calculating

$$p(\mathbf{x}) = \mathbb{E}_{\mathbf{z} \sim q_\phi(\mathbf{z}|\mathbf{x})}\left[ p_\theta(\mathbf{x}|\mathbf{z}) \frac{p_\theta(\mathbf{z})}{q_\phi(\mathbf{z}|\mathbf{x})} \right].$$

Here

- $q_\phi(\mathbf{z}|\mathbf{x})$ is a sample from a multivariate normal with mean and variance determined by the encoder
- $p_\theta(\mathbf{x}|\mathbf{z})$ is the probability of generating a given sequence from the decoded version of $\mathbf{z}$: a product of categorical probabilities
- $p_\theta(\mathbf{z})$ is the prior on the latent space

We found that 100 iterations of importance sampling yielded stable $P_{\mathrm{VAE}}$ estimates for our data, but used 500 iterations in the results presented here to ensure convergence.

## OLGA and selection model

We used OLGA (*Sethna et al., 2018*) with its default model parameters to evaluate recombination probabilities. We layered a selection model on top of this recombination model via a multiplicative factor $Q$, parameterized in terms of triples consisting of V gene identity, J gene identity, and CDR3 length. The roughly 14,000 parameters of this selection model $Q$ were estimated from the same training data used to train the VAE in each case. As derived in the supplementary material of *Elhanati et al. (2014)*, the maximum likelihood estimate of $Q$ for a given triple is the ratio of the empirical frequency of the triple in the data to the probability of observing the triple based on the recombination model. We truncated this ratio at 100 for numerical stability. We then used rejection sampling to sample from the corresponding $P_{\mathrm{OLGA.Q}}$ distribution. Code for estimating the `OLGA.Q` model parameters is included in our software package.

## Implementation and pipeline

We implemented our models in a modular fashion with extensive comments so that others can understand, reproduce, and build upon our work. Code and pipelines are available at https://github.com/matsengrp/vampire/ (*Matsen, 2019a*; copy archived at https://github.com/elifesciences-publications/vampire), while scripts and Jupyter notebooks (*Kluyver et al., 2016*) specific to this paper are available at https://github.com/matsengrp/vampire-analysis-1/ (*Matsen, 2019b*; copy archived at https://github.com/elifesciences-publications/vampire-analysis-1). All models were implemented in Python 3.6 using Keras 2.2.4 (*Chollet, 2015*) and the Tensorflow 1.11.0 backend (*Abadi et al., 2015*). Our pipeline is written with SCons (https://scons.org) and nestly (https://pythonhosted.org/nestly/; *McCoy et al., 2013*). The sumrep package depends heavily on the Immcantation framework (https://immcantation.readthedocs.io/; *Gupta et al., 2015*).

The following tools were also especially helpful:

- Biopython (*Cock et al., 2009*)
- cowplot (*Wilke, 2018*)
- ggplot2 (*Wickham, 2016*)
- GNU parallel (*Tange, 2018*)
- pandas (*McKinney, 2010*)
- scikit-learn (*Pedregosa et al., 2011*).

## Acknowledgements

We would like to thank Sam Sinai and Cheng Zhang for helpful discussions, Thierry Mora, Aleks Walczak, and Zachary Sethna for assistance with OLGA and the Q model, the AIRR software working group for their contributions to sumrep, Fred Hutch scientific computing, especially Michael Gutteridge and Dirk Petersen, and Adaptive Biotechnologies for hosting and sharing TCR data.

## Additional information

### Funding

| Funder | Grant reference number | Author |
|---|---|---|
| National Institutes of Health | R01 GM113246 | Kristian Davidsen<br>Branden J Olson<br>William S DeWitt III<br>Frederick A Matsen IV |
| National Institutes of Health | U19 AI117891 | Frederick A Matsen IV |
| National Institutes of Health | R01 AI120961 | Elias Harkins<br>Frederick A Matsen IV |
| Howard Hughes Medical Institute | Faculty Scholar grant | Kristian Davidsen<br>Branden J Olson<br>William S DeWitt III<br>Frederick A Matsen IV |
| National Institutes of Health | R01 AI146028 | Branden J Olson<br>Elias Harkins<br>Frederick A Matsen IV |
| National Institutes of Health | 5T32HG000035-23 | William S DeWitt III |

The funders had no role in study design, data collection and interpretation, or the decision to submit the work for publication.

### Author contributions

Kristian Davidsen, Conceptualization, Software, Formal analysis, Methodology, Writing—original draft, Writing—review and editing; Branden J Olson, Conceptualization, Software, Formal analysis, Visualization, Methodology, Writing—original draft, Writing—review and editing; William S DeWitt III, Jean Feng, Conceptualization, Software, Methodology, Writing—original draft, Writing—review and editing; Elias Harkins, Software, Methodology; Philip Bradley, Conceptualization, Software, Formal analysis, Supervision, Methodology, Writing—original draft, Writing—review and editing; Frederick A Matsen IV, Conceptualization, Data curation, Software, Formal analysis, Supervision, Funding acquisition, Visualization, Methodology, Writing—original draft, Writing—review and editing

### Author ORCIDs

Kristian Davidsen https://orcid.org/0000-0002-3821-6902
Branden J Olson https://orcid.org/0000-0003-1951-8822
William S DeWitt III https://orcid.org/0000-0002-6802-9139
Jean Feng http://orcid.org/0000-0003-2041-3104
Philip Bradley http://orcid.org/0000-0002-0224-6464
Frederick A Matsen IV https://orcid.org/0000-0003-0607-6025

### Decision letter and Author response

Decision letter https://doi.org/10.7554/eLife.46935.029
Author response https://doi.org/10.7554/eLife.46935.030

# Additional files

## Supplementary files

• Transparent reporting form
DOI: https://doi.org/10.7554/eLife.46935.019

## Data availability

Raw data (TCR sequences) is available at immuneACCESS: https://clients.adaptivebiotech.com/pub/emerson-2013-jim, https://clients.adaptivebiotech.com/pub/emerson-2017-natgen, https://clients.adaptivebiotech.com/pub/seshadri-2018-journalofimmunology, https://clients.adaptivebiotech.com/pub/deneuter-2018-cmvserostatus. Processed data is available through Zenodo: https://zenodo.org/record/2619576#.XKElTrfYphE. Code and instructions for reproducing figures is available at: https://github.com/matsengrp/vampire-analysis-1 (copy archived at https://github.com/elifesciences-publications/vampire-analysis-1). Code to process data and run VAE is available at: https://github.com/matsengrp/vampire/ (copy archived at https://github.com/elifesciences-publications/vampire).

The following previously published datasets were used:

| Author(s) | Year | Dataset title | Dataset URL | Database and Identifier |
|---|---|---|---|---|
| Emerson R, Sherwood A, Desmarais C, Malhotra S, Phippard D, Robins H | 2013 | Estimating the ratio of CD4+ to CD8+ T cells using high-throughput sequence data | https://doi.org/10.21417/B7H01M | immuneACCESS, 10.21417/B7H01M |
| Emerson R, DeWitt W, Vignali M, Gravley J, Hu J, Osborne E, Desmarais C, Klinger M, Carlson C, Hansen J, Rieder M, Robins H | 2017 | Immunosequencing identifies signatures of cytomegalovirus exposure history and HLA-mediated effects on the T-cell repertoire | https://doi.org/10.21417/B7001Z | immuneACCESS, 10.21417/B7001Z |
| DeWitt WS, Yu KKQ, Wilburn DB, Sherwood A, Vignali M, Day CL, Scriba TJ, Robins HS, Swanson WJ, Emerson RO, Bradley PH, Seshadri C | 2018 | A diverse lipid antigen-specific T cell receptor repertoire is clonally expanded during active tuberculosis | https://doi.org/10.21417/B7QG66 | immuneACCESS, 10.21417/B7QG66 |
| De Neuter N, Bartholomeus E, Elias G, Keersmaekers N, Suls A, Jansens H, Smits E, Hens N, Beutels P, Van Damme P, Mortier G, Van Tendeloo V, Laukens K, Meysman P, Ogunjimi B | 2018 | Memory CD4+ T cell receptor repertoire data mining as a tool for identifying cytomegalovirus serostatus | https://doi.org/10.21417/B7R91W | immuneACCESS, 10.21417/B7R91W |

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
