## [Decision Letter]

Thank you for submitting your article "Deep generative models for T cell receptor protein sequences" for consideration by *eLife*. Your article has been reviewed by two peer reviewers, and the evaluation has been overseen by Arup Chakraborty as the Senior and Reviewing Editor. The following individuals involved in review of your submission have agreed to reveal their identity: Eric Huseby (Reviewer #1); Curtis Callan (Reviewer #2).

The reviewers have discussed the reviews with one another and the Reviewing Editor has drafted this decision to help you prepare a revised submission.

Summary:

The paper presents a method, based on a certain generic machine learning protocol, for using T cell receptor (TCR) sequence data to capture the probability distribution from which this sequence data is drawn. This is a nontrivial thing to do, since one can easily convince oneself that actual data is sparsely drawn from the underlying, very high-entropy, distribution on sequence space. To capture the true distribution from sparse data, one must make restrictive assumptions on the form of that distribution, and the heart of the machine learning protocol is the assumption that the distributions to be learned are Gaussian (or Bernoulli). This is a very restrictive assumption, but it apparently works very well in a number of contexts, such as computer vision. You ask whether this approach can capture the diversity of the immune system, and present evidence that the answer is "yes". Therefore, we think this is a valuable contribution to our developing quantitative understanding of the stochastic nature of the adaptive immune system. We have concerns about the readability of the presentation (for the non-experts in machine learning), some technical issues, and some aspects of the way in which you present the significance of their work. These are noted below.

Essential revisions:

1) It would be helpful (in the Introduction, or at the relevant places in the Results) to describe the current understanding of the primary events of VDJ recombination. On the same line of discussion, it would be helpful to describe what events have been learned, and articulate clearly to a broad audience why this particular package is better than previous attempts to model TCR repertoires.

2) Few of the intended readers will be familiar with the Kingma and Welling (KW) approach and the conceptual context of that approach needs to be explained in much more concrete detail. To start with it has to be said explicitly that the encoder between data x and latent variables z is an explicit parametrized (Gaussian we think) probability distribution; similarly, it has to be said that the decoder from z to data variables x is also an explicit parametrized distribution (Bernoulli we think). It also has to be said that the prior on latent variables z is a very special distribution (isotropic unit variance on the z variable space, we think). Then it has to be said that the neural nets behind the encoder and decoder are actually maps from variables like x to the parameters of the various parametric distributions. As another specific clarity issue, the dimension of the latent space is not explicitly stated up front; one has to wait until somewhere deep in Materials and methods to realize that 20 is the chosen value (there is no discussion of why 20 as opposed to 200). More generally, There should be some discussion of why the KW method (statistics are Gaussian) has any reason to work in the context of understanding TCR statistics. An effort to rewrite the exposition so as to convey the conceptual heart of the method more clearly and explicitly would greatly enhance the utility of the paper to the quantitative biology readership.

3) It is argued that the VAE models can predict cohort frequencies at an R^2^ value of ~0.45, whereas a previous OLGA model works at an R^2^ value of ~0.25. In absolute terms, the VAE model is better, however, neither works particularly well; both have R^2^ values less than a minimal cutoff of 0.5. Much more clarity is required in describing the comparison between the two methods.

OLGA is a different way of capturing TCR sequence statistics, and it relies on the idea that there are biological hidden variables (associated with the VDJ recombination process) whose statistics can be inferred from sequence data and then these statistics used to compute probabilities of finding individual TCR sequences in new data. OLGA doesn't include selection effects that happen post-VDJ recombination and that shape the statistics of observed in-frame TCR sequences. This is where the Q in OLGA.Q comes in. You construct a version of a selection model which is too simplistic to do a good job of capturing selection effects. Comparing the results of their machine learning approach to OLGA.Q to say that former captures "more" aspects of TCR sequence statistics than the latter doesn't seem very appropriate. Are TRAV rearrangements considered? TRBV/TRBJ rearrangement and usage is largely ignorant of selection context. TCRb rearrangement occurs prior to TCRa rearrangement and its usage is largely ligand independent. TCRa rearrangement is later, and subject to strong selective pressures. In particular, if TCRa rearrangement produces a non-signaling TCR, TCRa rearrangement occurs again with more 3' gene segments. Control mechanisms and the "rules" that govern TCRa processes are much less well understood. These aspects are not accounted for, and should be stated.

4) Because it assumes that Gaussian core distributions underlie the observed data, it is by no means obvious that the Kingma and Welling method is appropriate for TCR data. The most attractive feature of the KW approach is that it provides a method for computing the intrinsic probability of finding any specific TCR clone in a new data sample (what the authors call P_VAE_). The paper shows histograms of this quantity over various data sets, and these histograms have the striking feature that the probability values start small and range over more than ten orders of magnitude. Now the OLGA method, taking a totally different approach, can also calculate the one-shot probability (called P_gen_) of any specific clone being created in a single VDJ recombination event and one can plot the same sort of histogram of generation probabilities. What is interesting is that the two approaches produce very similar generation probability histograms. The further modifications to OLGA predictions due to selection might change probabilities by modest factors, but we are talking about probabilities that range over more than ten orders of magnitude, so the approximate compatibility of the two methods is evidence that the KW approach is doing a good job of capturing the stochastic effects of selection. The relatively poor fit of both methods may also reflect TCR sequencing errors (PCR amplification and sample limited constraints) as well as the more significant problem alluded to in the end of the Discussion. That is, how to deal with the error associated with point estimates of low and very low frequency TCR rearrangements. This is illustrated in Figure 2 by the triangle shape of the log-log plots, where there is a greater level of divergence at low versus high frequency TCRs. Some discussion or accounting for these issues would be help to the reader.

---

## [Author Response]

Essential revisions:1) It would be helpful (in the Introduction, or at the relevant places in the Results) to describe the current understanding of the primary events of VDJ recombination.

We have added a condensed introduction to V(D)J recombination in the first paragraph of the Introduction:

“T cell receptors (TCRs) are composed of an *α* and a *β* protein chain, both originating from a random V(D)J recombination process, followed by selective steps that ensure functionality and limit auto-reactivity. […] The naive T cell population consists of T cells that have undergone V(D)J recombination and MHC selection but not yet encountered antigen.”

This includes important information about the order of the events (noted in point 3 of these comments). We also now more consistently refer to V(D)J when referring to the general process, but use VDJ when discussing the *β* chain specifically.

On the same line of discussion, it would be helpful to describe what events have been learned, and articulate clearly to a broad audience why this particular package is better than previous attempts to model TCR repertoires.

We have now added the following material to the Discussion adding to our description of what has been learned.

“Specifically, we use a semiparametric method that makes a single weak assumption: that there exists some small number of latent parameters that can be used to generate to the observed distribution. […] The basic model, for example, is implemented in about 100 lines of Python code.”

We have also added another lesson learned:

“Furthermore, our efforts to inject biological knowledge into the deep learning framework did not significantly improve performance.”

2) Few of the intended readers will be familiar with the Kingma and Welling (KW) approach and the conceptual context of that approach needs to be explained in much more concrete detail. To start with it has to be said explicitly that the encoder between data x and latent variables z is an explicit parametrized (Gaussian we think) probability distribution; similarly, it has to be said that the decoder from z to data variables x is also an explicit parametrized distribution (Bernoulli we think). It also has to be said that the prior on latent variables z is a very special distribution (isotropic unit variance on the z variable space, we think). Then it has to be said that the neural nets behind the encoder and decoder are actually maps from variables like x to the parameters of the various parametric distributions. As another specific clarity issue, the dimension of the latent space is not explicitly stated up front; one has to wait until somewhere deep in Materials and methods to realize that 20 is the chosen value (there is no discussion of why 20 as opposed to 200). More generally, There should be some discussion of why the KW method (statistics are Gaussian) has any reason to work in the context of understanding TCR statistics. An effort to rewrite the exposition so as to convey the conceptual heart of the method more clearly and explicitly would greatly enhance the utility of the paper to the quantitative biology readership.

Thank you for these comments. We have now provided additional detail about the operation of the encoder and decoder, specifically calling out the distributions used:

“In this paper the latent space is 20-dimensional, and we use the conventional choice of a standard multivariate normal prior for *p_θ_*(**z**). […] The decoder *p_θ_***x̂|z**) is a per-site categorical distribution over amino acids and gaps parameterized by a neural network with input **z**.”

We also describe why we thought that VAEs could be useful in the TCR context: “Previous work using VAEs have found success when first, there is a vast amount of data available, and second, the data distribution is complicated, involving nonlinearities and interactions between covariates. There is indeed a vast amount of TCR repertoire data, and the TCR probability distributions are complex.”

We also clarify that the choice of a normal distribution is not a classical “model choice” based on a mental model of the underlying biology, but rather out of mathematical convenience:

“This choice of a normal distribution is primarily for mathematical convenience rather than being part of a specific modeling design; the normal “noise” in the latent space get processed by a neural network which introduces non-linearities that ensure that the result is not normal. However, VAE variants do use other distributions in place of normal (Dilokthanakul et al., 2016; Davidson et al., 2018).” This text also emphasizes that the neural network offers substantial flexibility, transforming the normal with non-linearities.

As described above, the conceptual heart of the method is simply that TCR repertoire distributions can be generated by latent distributions on a relatively small number of parameters. These methods have tremendous flexibility and generality: given the vast amount of TCR data now available, one can learn models that capture features of the distributions that we do not yet even conceptualize. Indeed, part of the message of the paper is that even very simple off-the-shelf neural network methods can compete with refined classical models, and that future developments on neural network design and training will enable much better models.”

Regarding 20 vs. 200 dimensions of the latent space, we have now added the following material:

“Model design and parameter tuning, including the sizes of hidden layers and the dimension of the latent space, was performed using the data of DeWitt et al., 2018). On this data we endeavored to decrease model size without incurring loss on held-out data within this data set. We found that the model was relatively robust to parameter perturbations as long as the number of parameters was not too small.”

We welcome any suggestions concerning how we can make these points more clear.

3) It is argued that the VAE models can predict cohort frequencies at an R^2^ value of ~0.45, whereas a previous OLGA model works at an R^2^ value of ~0.25. In absolute terms, the VAE model is better, however, neither works particularly well; both have R^2^ values less than a minimal cutoff of 0.5. Much more clarity is required in describing the comparison between the two methods.

We have tamped down the “high accuracy” claims. However, we do feel that these R^2^ values show that our VAE has predictive power. Recall that R^2^ can be interpreted as the proportion of variance explained. Although there is room for improvement, we feel that getting almost 50% variance explained for an out-of-sample frequency prediction applying a biology-agnostic model to just amino acid sequence across many orders of magnitudes of frequency is a significant achievement.

As suggested below, we have added two sentences describing additional challenges with the data we have at hand:

“Recall that these correlation measures include the full scale of frequencies, including very noisy frequency estimates on the lower end of the scale. Also, we make no efforts to account for sequencing error above the methods used in DeWitt et al., 2018.”

Regarding clarity of comparison description, see below and note that we will be opening up our analysis repository upon publication. This repository is composed of a series of Jupyter notebooks performing the analysis reproducibly.

OLGA is a different way of capturing TCR sequence statistics, and it relies on the idea that there are biological hidden variables (associated with the VDJ recombination process) whose statistics can be inferred from sequence data and then these statistics used to compute probabilities of finding individual TCR sequences in new data. OLGA doesn't include selection effects that happen post-VDJ recombination and that shape the statistics of observed in-frame TCR sequences. This is where the Q in OLGA.Q comes in. You construct a version of a selection model which is too simplistic to do a good job of capturing selection effects. Comparing the results of their machine learning approach to OLGA.Q to say that former captures "more" aspects of TCR sequence statistics than the latter doesn't seem very appropriate.

This is an important point that is addressed with the following sentences that introduce our version of the OLGA.Q model:

“This is a simpler model than the general Elhanati et al., 2014 model, which allows for selection based on CDR3 amino acid composition. […] In any case, an implementation of the general Elhanati et al., 2014 model, for which training is highly involved, is not currently available.”

Specifically, our model is richer than any of the models currently in use, including models from the same group of Elhanati et al., 2014, that model the underlying frequency of TCRs in the functional repertoire. We welcome further suggestions about how we can make this point more clear.

We also note that we have implemented the first publicly-available method for sampling sequences from the Ppost distribution of OLGA.Q.

Are TRAV rearrangements considered? TRBV/TRBJ rearrangement and usage is largely ignorant of selection context. TCRb rearrangement occurs prior to TCRa rearrangement and its usage is largely ligand independent. TCRa rearrangement is later, and subject to strong selective pressures. In particular, if TCRa rearrangement produces a non-signaling TCR, TCRa rearrangement occurs again with more 3' gene segments. Control mechanisms and the "rules" that govern TCRa processes are much less well understood. These aspects are not accounted for, and should be stated.

We do not consider TRAV in this paper, however, it is a very interesting topic for future work as we now note:

“Here we have restricted our attention to TCR *β* sequencing, however, our methods apply with no modification to TCR *α* chains. […] The most interesting insights will come from jointly modeling the two chains using large-scale *αβ* paired TCR sequencing (Howie et al., 2015), which is a more complex process.”

4) Because it assumes that Gaussian core distributions underlie the observed data, it is by no means obvious that the Kingma and Welling method is appropriate for TCR data. The most attractive feature of the KW approach is that it provides a method for computing the intrinsic probability of finding any specific TCR clone in a new data sample (what the authors call P_VAE_). The paper shows histograms of this quantity over various data sets, and these histograms have the striking feature that the probability values start small and range over more than ten orders of magnitude. Now the OLGA method, taking a totally different approach, can also calculate the one-shot probability (called P_gen_) of any specific clone being created in a single VDJ recombination event and one can plot the same sort of histogram of generation probabilities. What is interesting is that the two approaches produce very similar generation probability histograms. The further modifications to OLGA predictions due to selection might change probabilities by modest factors, but we are talking about probabilities that range over more than ten orders of magnitude, so the approximate compatibility of the two methods is evidence that the KW approach is doing a good job of capturing the stochastic effects of selection. The relatively poor fit of both methods may also reflect TCR sequencing errors (PCR amplification and sample limited constraints) as well as the more significant problem alluded to in the end of the Discussion. That is, how to deal with the error associated with point estimates of low and very low frequency TCR rearrangements. This is illustrated in Figure 2 by the triangle shape of the log-log plots, where there is a greater level of divergence at low versus high frequency TCRs. Some discussion or accounting for these issues would be help to the reader.

Regarding Gaussian distributions underlying the observed data, we hope it is clear from our response above that this is a Gaussian on the latent space before application of the neural network. A Gaussian transformed by a neural network is not Gaussian in general, and can be quite “far” from a Gaussian using even simple neural network transformations.

We now emphasize the difficulty posed by low frequency rearrangements or sequencing error when we introduce the R^2^ results, as described above.